# Monetary Policy Implications on Macroeconomic Performance in the Common Monetary Area: A Panel-SVAR Framework

**Theron Shumba *** and **Sophia Mukorera**

School of Accounting, Economics and Finance, University of KwaZulu-Natal, King Edward Ave, Scottsville, Pietermaritzburg 3209, South Africa

\* Correspondence: theronshumba@gmail.com or 214585025@stu.ukzn.ac.za

**Abstract:** The CMA (Common Monetary Area) is a quadrilateral monetary arrangement encompassing South Africa, Namibia, Lesotho, and Eswatini. The four countries have undergone a gradual improvement in regional economic integration for the effective economic coordination of their policymaking. Despite the monetary coordination, the countries are still experiencing poor economic performance. This study traces how a shock or an unanticipated change in the anchor country's central bank's policy instrument, in this case, South Africa, affects the macroeconomic performance in the entire CMA region. Employing a Panel Structural Vector Autoregressive model (Panel-SVAR) and annual data from 1980–2021, the findings show that a positive shock in the repo rate from South Africa significantly affected important macroeconomic performance indicators. The results indicate that a shock in the anchor country's repo rate is followed by a significant decline in RGDP_G, a decrease in inflation, a decrease in money supply, and an increase in lending rate in the entire CMA region. The study recommends that CMA monetary authorities and policymakers need to formulate policies toward cushioning the effects of unanticipated monetary policy shock from the anchor country as well as global shocks.

**Keywords:** panel-structural vector autoregressive; monetary policy shocks; common monetary area

## 1. Introduction

The CMA is a quadrilateral monetary arrangement made up of Namibia, South Africa, Lesotho, and Eswatini (LENS). The main aim of its establishment is to enhance development among participant economies and effectively foster the coordination of economic policies (Wörgötter and Brixiova 2019). This agreement places South Africa as the domineering or anchor country, which sets the economic pace for other participant countries. These small member countries submit their exchange rate policies to the South African Reserve Bank (SARB), hence, a fixed exchange rate system of the South African Rand against the Lesotho Loti, Namibian Dollar, and Eswatini Lilangeni on a one-to-one basis was established (Seleteng 2016). The CMA agreement restrains small member countries: Namibia, Eswatini, and Lesotho (LEN), from exercising total discretion in monetary policy formulation and implementation (Seoela 2020). At the same time, South Africa expresses total dominance when formulating the monetary policy of the entire region (Nagar 2020). However, South Africa (SA) primarily targets its own economy when executing monetary policy (Kamati 2014). The implications of the anchor country's monetary policy decisions are deemed to be conveyed to all participant members (Kamati 2020).

Monetary policy is a vital tool utilised for stimulating growth, reducing unemployment, and achieving stable prices in an economy through the monetary transmission mechanism (MTM) process. According to Nagar (2020), as much as monetary policy is a critical instrument for improving the performance of an economy and stabilising prices, it is vulnerable to shocks which can be detrimental to economic goals.

In the CMA region, the empirical evidence portrays that despite the regional collaboration, the countries are still experiencing poor economic growth (in terms of real GDP).

For example, in 2020, on average, the RGDP_G of the CMA region declined to −6.91%, individually, Lesotho's real GDP growth rate decreased to −11.06%, Namibia's to −7.98%, Eswatini's to −1.64% and South Africa's to −6.96% (World Bank Development Indicators 2022). Poor economic growth discourages firms from investing; businesses become unwilling to hire workers, and consumer spending becomes low, leading to low productivity, high price levels, job losses, or high unemployment (Nagar 2020).

In such a case, monetary policy authorities employ expansionary or loose monetary policy that lowers the interest rate or indirectly increases the money supply to stimulate the economy (Nagar 2020). In the literature, the traditional Keynesian theory suggests that an easing of monetary policy decreases the cost of borrowing or lowers the interest rate, which induces higher investment spending and more growth in output (Mishkin 1995). Moreover, Kamati (2020) asserted that in the CMA, the Taylor rule is instrumental in explaining how South African monetary authorities regulate the interest rate to stimulate economic performance.

Monetary policy shock is defined as an unanticipated variation in a monetary tool, such as money supply or short-term interest rate, that accounts for changes in prices or output (Nielsen et al. 2005). Monetary policy shocks can result in a positive or negative impact on economic performance. For instance, the study of Kamati (2014), using an SVAR framework in Namibia, revealed that a positive monetary policy shock resulted in a decline in output, a fall in inflation, and a decrease in credit. However, although Namibia is a member of the CMA, the study did not examine the impact of the monetary policy shock in the CMA region. Moreover, Cheng (2006), using an SVAR econometric model, investigated the effect of a shock on prices, output, and exchange rate in Kenya from 1997 to 2005. The findings revealed that a tight monetary policy was followed by a price decrease, with an insignificant effect on output.

According to the general equilibrium theory, when a shock occurs, the economy adjusts itself to restore the economy to equilibrium creating a business cycle (Ennis 2018). Ennis (2018) asserted that movements in business cycles could be compared with trends in the short-term interest rate. Their argument stems from the view that the interest rate is a useful monetary policy instrument for inducing appropriate adjustment policies for economic stabilisation.

The trend analysis of the movement of the business cycles and short-term interest rates of CMA economies (Figures 1–4) shows that only the RGDP_G of South Africa responded to the changes in short-term interest rates in line with the general theoretical expectations. Other member countries' reactions to domestic short-term interest rate changes reflected unconventional responses. South Africa implemented an expansionary monetary policy through a decline in interest rates during recession periods and a restrictive monetary policy during periods of an economic boom. However, in other member countries, Figures 1–4 shows the opposite. For instance, Eswatini experienced a recession from 1981 to 1983, yet its curve indicates a restrictive monetary policy was implemented. The movement in the business cycle of Lesotho in periods of recession (1987–1989), shows that a tight monetary policy was implemented instead of a loose monetary policy. Moreso, Namibia also experienced a boom between 2009 and 2010, but evidence from the business cycle movement shows that a loose monetary policy was instituted.

Moreover, despite introducing the inflation target framework by the anchor country in 2000, some CMA members have been experiencing an inflation rate beyond the target bandwidth of 3–6%. For instance, Lesotho reached an all-time high inflation rate of 35.15% in 2002 (World Bank Development Indicators 2022). We hypothesise that shocks from South Africa might have adverse spillover effects on the performance indicators of CMA countries. The empirical observations in Figures 1–4 reveal that despite the collaboration of CMA countries, the region poses a high vulnerability to spillover effects of monetary policy shocks from the anchor country, which could make their stabilising domestic policies ineffective. Therefore, this study fills the literature gap by investigating the impact of

monetary policy shocks from the anchor country on performance variables such as output growth and inflation in the CMA region.

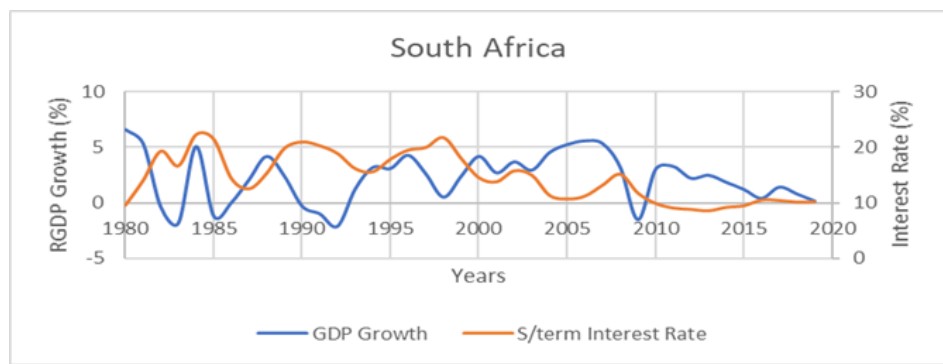

**Figure 1.** Movement in South Africa's business cycle. Source: Computations by the author using data from Stats SA (2022) and World Bank Development Indicators (2022).

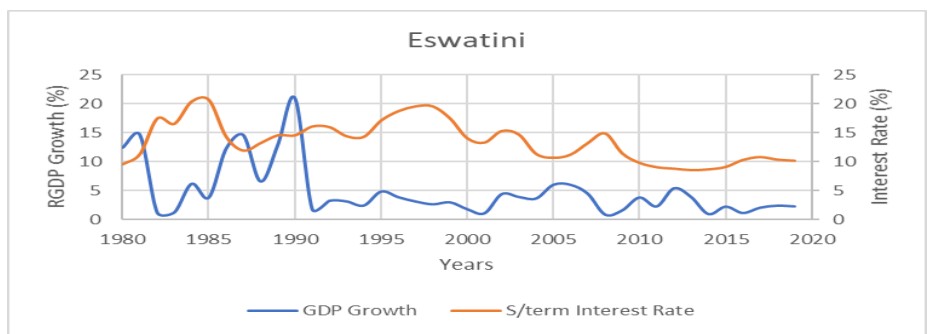

**Figure 2.** Movement in Eswatini's business cycle. Source: Computations by the author using data from Central Bank of Eswatini (2022) and World Bank Development Indicators (2022).

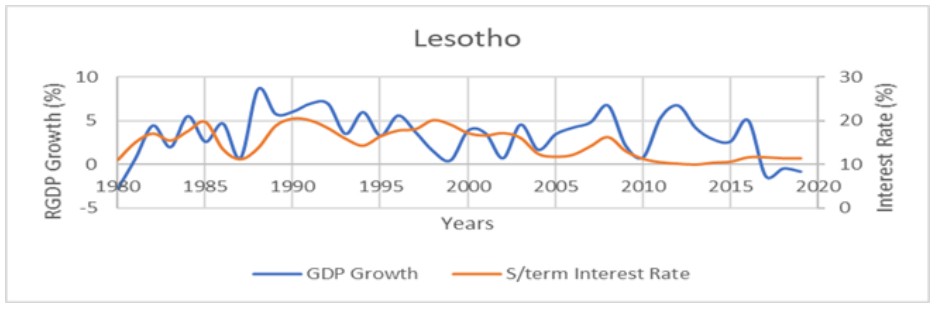

**Figure 3.** Movement in Lesotho's business cycle. Source: Computations by the author using data from the Central Bank of Lesotho (2022) and World Bank Development Indicators (2022).

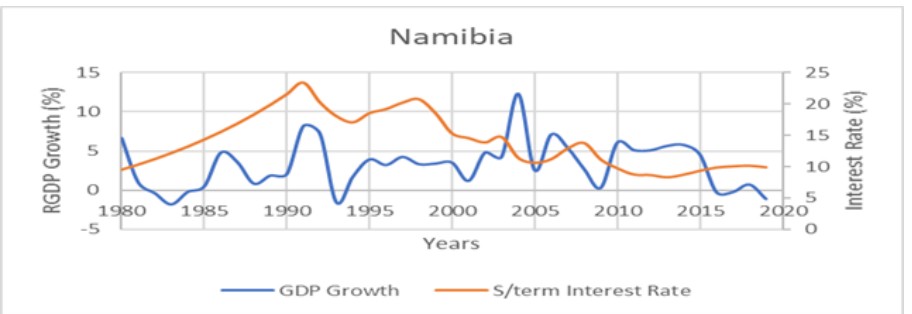

**Figure 4.** Movement in Namibia's business cycle. Source: Computations by the author using data from the Bank of Namibia (2022) and World Bank Development Indicators (2022).

Conclusively, past studies in South Africa reveal that periods of restrictive monetary policy have a negative effect on output growth in South Africa, specifically industrial output (Kutu and Ngalawa 2016; Kabundi and Ngwenya 2011; Seleteng 2016; Ikhide and Uanguta 2010). However, very few studies have extended their analysis to the CMA region. There is still a lack of clarity on how monetary decisions from South Africa can influence the CMA region and the transmission channels through which the effects are conveyed. The objective of the study is focused on evaluating monetary-policy-induced innovations and how they affect macroeconomic performance indicators in the region. This study closes the literature gap by comparatively analysing the transmission of monetary policy shocks in the CMA.

## 2. Economic Performance in CMA

Economic performance can be defined as a measurement or indicator of how well an economy is doing in achieving its most crucial objectives (Kashima 2017). According to Chileshe et al. (2018) the most predominant and key economic objectives targeted by any economy are price stability and a stable high rate of economic growth. The main objective behind the establishment of the CMA monetary union is in fostering economic advancement and development of less developed participant members (Masha et al. 2007). In the measurement of economic performance economic growth, the Real Gross Domestic Product Growth (RGDP_G) rate has been regarded as a prominent indicator in the CMA region (Kashima 2017). RGDP_G reveals the annual percentage change in the total value of goods and services produced in an economy adjusted for inflation (Jane et al. 2018).

As evidenced by the graphical presentation in Figure 5, the RGDP_G for all Common Monetary Area participants demonstrates close association from 1980 to 2021 since they trend together between these periods. The trend line depicts a below 5% growth rate, showing that the region has not been experiencing positive growth on average. The sharp peaks in the co-movement (for example in 1990 Eswatini reached an all-time highest growth rate of 21.01%, and in 2004 Namibia recorded a sharp peak of 12.27%) were attributed to the fact that both economies' growth rate was boosted by surging exports (Kashima 2017). In Namibia the lowest growth rate of −1.58% in 1993 and −1.25% for Lesotho in 2009, despite the close economic ties with South Africa, were due to the shift in some foreign investors who moved from these countries to invest in South Africa directly because of close proximity to South African markets (Seleteng 2016). In 2019 the contraction and decline in the growth rate of all CMA economies could be attributed to strict lockdown measures, which constrained the trade and adversely affected the financial markets of CMA economies. However, a rebound in the growth rate of CMA economies from 2021 could be explained by the easing of lock down restrictions in support of export-oriented sectors (Yingi 2022). In addition, another variable incorporated to measure economic performance is inflation. Seleteng (2016) asserted that the CMA seems to have benefited from low inflation rates after the anchor country's inflation target framework of 2000 was introduced. However, there are periods after the CMA agreement of 1986 which reveal that inflation became more volatile in CMA countries. For instance, Eswatini recorded an all-time highest figure of 31.14% in 1988, Namibia hit its all-time highest inflation rate of 20.56% in 1992 after officially joining the CMA, Lesotho recorded an all-time highest figure of 35.15% in 2002, in the same year Namibia reached 12.72%, and in 2008 Namibia's inflation rate was annualised at 9.7% (World Bank Development Indicators 2022).

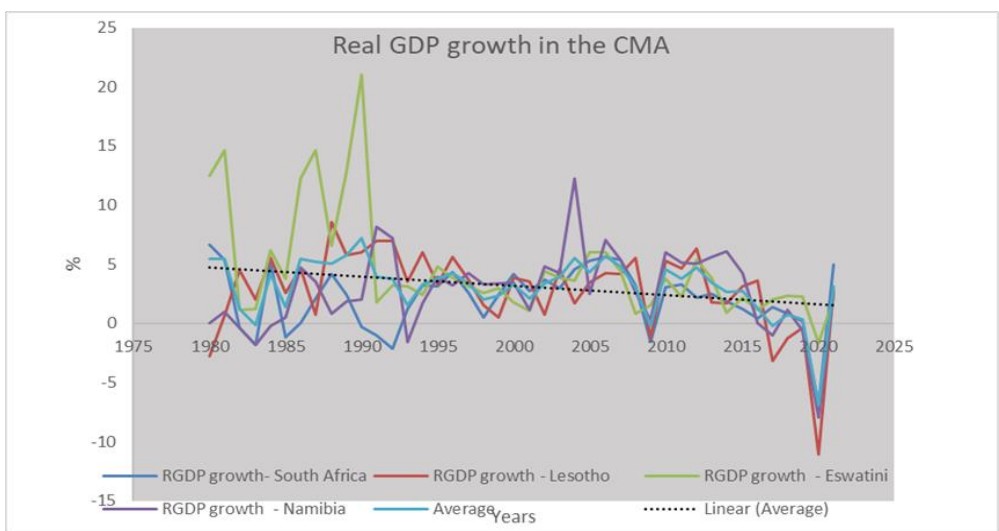

**Figure 5.** Economic Growth trends of CMA economies from 1980–2021. Source: Author's Computations using data from World Bank Development Indicators (2022).

Furthermore, as depicted in Figure 6 below, the CMA countries' lending rates from 1980 to 2021 also show strong co-movements in the direction of the South African repo rate from 1980. The explanation of these co-movements is because the short-term interest rates in all member countries are set close to the repo rate of South Africa in order to maintain the agreed fixed peg.

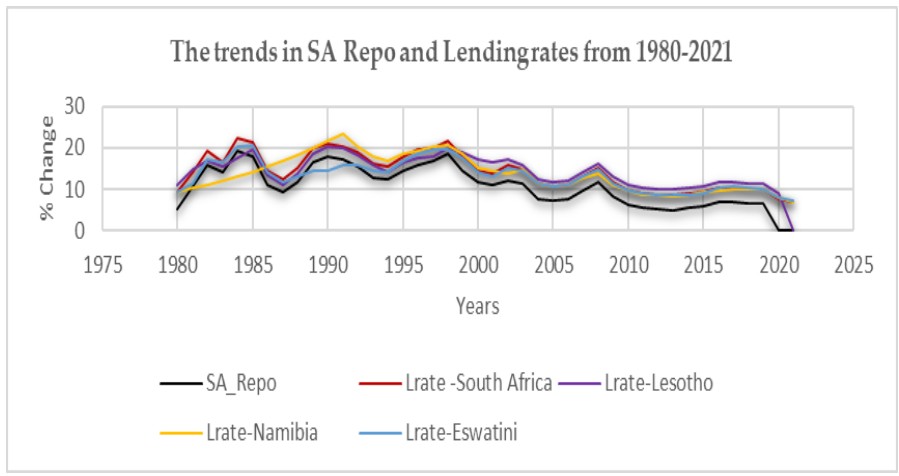

**Figure 6.** CMA Trends in Lending rates and SA Repo. Source: Computations from World Bank Development Indicators (2022).

## 3. Monetary Policy Shocks and Economic Performance

The Taylor rule and the traditional Keynesian theory forms the theoretical foundational basis for investigating the implications of the transmission of a shock from South Africa across the CMA region. According to Rossouw and Padayachee (2020), the Taylor rule propounds how a 1% increase in inflation is supposed to propel the central bank of South Africa to raise the short-term interest rate by more than 1%. Whereas in the traditional Keynesian theory, unlike the classical theory, it is posited that the economy does not remain at full employment equilibrium. They believe disequilibrium emanates from economic instability (Mishkin 1995).

Furthermore, although this study is rooted in the Taylor rule and the traditional Keynesian theories on monetary policy, in the body of literature, empirical studies that examine the spillover effects of a shock from one economy to the economic performance of other

economies are very few. For instance, Mirdala (2009) investigated the monetary spillover effects using a structural VAR econometric model in the Visegrad Group countries (Slovak Republic, Hungary, Poland, and Czech Republic). The findings revealed that a shock strongly impacted the real output of Visegrad countries, while the effect on inflation was inconclusive among the countries. In some countries, the shock resulted in an increase in inflation, while in others, it decreased. Barigozzi et al. (2014), using pooled data, employed a structural dynamic factor model in North European and South European countries; their findings revealed that after an expansionary policy, there are some differences in response to inflation or prices in North and South Europe. Whereas Angeloni et al. (2003), using the SVAR framework, found that an increase in interest rates had no observable effect on prices in the initial stages, whilst output decreased temporarily.

The most notable study of Cavallo and Ribba (2015) using a structural (near) VAR investigated the impact of area-wide shocks with particular attention to monetary policy shocks. Their conclusion was that a contractionary monetary policy causes similar recessionary effects in all countries and that as far as business cycle fluctuations are concerned, the largest European economies were mainly explained by common area-wide shocks, whereas in the second category of economies comprising Portugal, Ireland, and Greece, national shocks played a greater role.

Moreso, Georgiadis (2016), employed a Global VAR (GVAR) model to assess the global spillovers from identified US monetary policy shocks. The study found out that US monetary policy generates sizable output spillovers to the rest of the world, which are larger than the domestic effects in the US for many economies. The results suggested that policy makers could mitigate their economies' vulnerability to US monetary policy by fostering trade integration as well as domestic financial market development, increasing the flexibility of exchange rates, and reducing frictions in labour markets.

Buigut (2009), estimated a three-variable recursive VAR for three East African Community (EAC) countries using data from 1984 and 2006. The paper found that a shock to the short-term interest rate was found to have no statistically significant effect on real output and inflation. Nevertheless, Bikai and Kenkouo (2015) argued that these findings are biased by the fact that the study used a sample that includes too few observations for empirical analyses, resulting in few degrees of freedom. Bikai and Kenkouo (2015) proposed that using a panel-SVAR model could resolve these issues because it provides an effective way of dealing with over-parameterisation. In contrast, their results found that an expansionary monetary increases prices significantly in Kenya and Uganda, while output increases in Burundi, Kenya, and Rwanda. Similar to the current study, there are very few studies that compare the effect of South Africa's monetary policy conduct on the performance of the CMA countries. Ikhide and Uanguta (2010) and Seleteng (2016) both examined the impact of SARB's monetary policy on the CMA economies using a VAR framework. Unlike the approach in this study, Ikhide and Uanguta (2010) used monthly data but excluded economic output from the estimated models, while Seleteng (2016) used annually aggregated data. These studies focused on how changes in the SARB's monetary policy instrument (repo rate) affect the money supply, credit, and prices in the CMA and thus evaluate the ability of the CMA economies to undertake independent monetary policy. Both studies found statistically significant results that lending rates and price levels were instantaneously sensitive to changes in the repo rate. However, Ikhide and Uanguta (2010) also found that money supply is instantaneously responsive to the repo rate, while Seleteng (2016) did not find any significant relationship.

Overall, the previous studies show opposing views on the impact of monetary policy on economic performance. Some support that increasing interest rates (contractionary monetary policy) negatively affects the economy, while others support that it has no observable impact on the economy.

However, the monetary interdependence in the CMA region between the leading economy (South Africa) and other member countries may create a high possibility of monetary spillover effects. This implies that the transmission of monetary policy shocks

from South Africa can potentially influence the performance of participant economies positively or negatively.

In the CMA, the study of Ikhide and Uanguta (2010) examined how innovations in the South African repo rate influence money, credit, and price levels while investigating the small member countries' capability to undertake independent monetary policy. Their findings revealed that the repo rate (SA_REPO) is a more effective monetary tool in influencing the participant countries in the CMA than bank rates. Ikhide and Uanguta (2010) found that other CMA countries could not independently undertake policies. Nevertheless, as mentioned in their study, the shortage of gross domestic product data was one of the challenges they encountered. A recent study by Seoela (2020) found that across the participant LENS countries, lending rate spread, credit, and money supply reacted asymmetrically to monetary innovations. However, Seoela's (2020) study's main drawback was that the author used the Chow–Lin approach of disaggregating data since output data were unavailable. The main challenge of such an approach is that it yields misleading results since the covariance matrix is unknown. Hence, this study fills the literature gap by employing annual real output growth data from 1980 to 2021. Another main challenge highlighted in the results of Seoela (2020) was the price puzzle problem. Seoela's (2020) findings reveal that inflation increases in some countries after a one-standard-deviation shock is introduced to the South African repo rate. This empirical anomaly identified in Seoela's (2020) study is referred as the price puzzle. Rossouw and Padayachee (2020) asserted that the price puzzle occurs when monetary policy committee of South Africa implement the short-term interest rate without sufficient observation of inflation signals in the future. Nielsen et al. (2005) argued that including commodity prices helps to tame the price puzzle problem. This study added to the body of knowledge by including commodity prices to purge and mitigate the price puzzle problem, which was not incorporated in the studies of Seoela (2020), Ikhide and Uanguta (2010), or Seleteng (2016).

Moreover, past studies reveal that there is no consensus about the impact of monetary policy shocks in the CMA region. Ikhide and Uanguta (2010) found out that a shock in the repo rate decreases prices or inflation in CMA countries, whereas the results of Dlamini (2018) and Seoela (2020) reflect the opposite.

## 4. Methodology

### 4.1. Theoretical Framework

In this study, the interest rate channel is the closest channel in investigating how a monetary shock is transmitted to the whole region from South Africa. Following Ikhide and Uanguta (2010), the theoretical framework to analyse the close connection among CMA members is stated as follows:

$$\uparrow Repo \Rightarrow rrates \ \uparrow \Rightarrow (Investment \downarrow C \downarrow) \Rightarrow \downarrow Y$$

A restrictive monetary policy action portrayed by a rise in the South African repurchase rate ($\uparrow Repo$) will in turn lead to a rise in other short-term interest rates depicted as ($\uparrow rrates$). The increase in the repo rate, shown above, spreads or is transmitted to other short-term interest rates raising the cost of borrowing. This high cost of capital or cost of borrowing provokes a fall in consumer ($\downarrow C$) and investment ($\downarrow I$) spending and ultimately reduces aggregate output ($\downarrow Y$).

Supposing that the CMA region is represented by the following structural equation:

$$EY_{it} = F_{i0} + B_1 Y_{it-1} + B_2 Y_{it-2} + \cdots + B_k Y_{it-k} + \Pi X_t + V \varepsilon_{it} \tag{1}$$

$E$ depicts an invertible ($f \times f$) matrix showing contemporaneous relationship among the variables; $Y_{it}$ represents a ($f \times 1$) vector of variables that our model has specified as endogenous such that $Y_{it} = Y_{1t}, Y_{2t}, Y_{3t} \ldots \ldots Y_{nt}$. $F_{i0}$ is a ($f \times 1$) vector reflecting the intercept terms of the countries; $i$ denotes variables specific for each country. $B_i$ is a ($f \times f$) matrix of coefficients of lagged endogenous variables (for every $j = 1 \ldots .k$); $\Pi$ and $X_t$ are

vectors of coefficients and the exogenous variables, respectively, capturing external shocks; $V$ is a $(f \times f)$ matrix whose non-zero diagonal elements allow for the direct impact of some shocks on other endogenous variables in the model. Finally, $\varepsilon_{it}$ represents a vector of white noise structural disturbances or uncorrelated error terms. According to Enders (2004), the $Panel - SVAR$ first Equation (1) representing the CMA region is inestimable. This is because of the feedback effects incorporated in the structure, which allow endogenous variables to influence each other in the past and current realisation time path of $EY_{it}$. However, the information in the system can be recovered, as suggested by Enders (2004), by transforming the structure into a reduced form $Panel - SVAR$.

The first Equation (1) is pre-multiplied by the inverse of $E$, which is $(E^{-1})$, which yields a reduced form Panel-SVAR model in Equation (2) such that

$$Y_{it} = E^{-1}F_{io} + E^{-1}B_1 Y_{it-1} + E^{-1}B_2 Y_{it-2} + \cdots E^{-1}B_k Y_{it-k} + E^{-1}\Pi X_t + E^{-1}V\varepsilon_{it} \quad (2)$$

Depicting,

$$E^{-1}F_{i0} = D_i, \ E^{-1}B_1 \ldots E^{-1}B_k = E_i \ldots \ldots E_k, \ E^{-1}\Pi = \alpha \text{ and } E^{-1}V\varepsilon_{it} = \mu_{it} \quad (3)$$

Hence Equation (3) becomes:

$$Y_{it} = D_i + E_1 Y_{it-1} + E_2 Y_{it-2} + \cdots E_k Y_{it-k} + \alpha X_t + \mu_{it} \quad (4)$$

Equation (1) differs from Equation (4) in the fact that the first is known as a primitive $Panel - SVAR$ where all variables affect each other in a contemporaneous way. The second equation is termed the reduced form $Panel - SVAR$ whereby all the variables on the right-hand side of the equation are pre-determined at a time $(t)$ and there is no variable which has an immediate or direct influence on another in the system. Moreover, the error term $\mu_{it}$ is a composite of shocks in $Y_{it}$ which represents a vector of endogenous variables for all CMA economies at a time $(t)$ (Enders 2004).

Equation (4) can be rewritten as follows:

$$Y_{it} = D_i + E(L)Y_{it} + K(L)X_t + \mu_{it} \ldots \quad (5)$$

where $X_t$ and $Y_{it}$ are $(n \times 1)$ vectors of exogenous and endogenous variables of CMA economies in that order, such that

$$X_t = (SA\_REPO, \ COMM\_PRICES) \quad (5a)$$

$$Y_{it} = (INF, MS, \ LRATE, \ RGDP\_G) \quad (5b)$$

Furthermore, $D_i$ encapsulates a vector of constants which represents the intercept terms of every CMA country, whereby $E(L)$ and $K(L)$ are matrices of polynomial lags which capture the behaviour between endogenous variables and its lags (Herve 2017). $\mu_{it} = E^{-1}V\varepsilon_{it}$ depicts a vector of uncorrelated error terms which can be re-arranged as $E\mu_{it} = V\varepsilon_{it}$.

The Equations (4) and (5) share similar features in line with Herve (2017), because they are both reduced form $Panel - SVARs$ derived from the primitive $Panel - SVAR$ framework of Equation (1).

Moreover, in order to get back the information in the structural equation we apply restrictions on matrix $E$ and matrix $V$ system of equations:

$$
\begin{bmatrix} 1 & 0 & 0 & 0 & 0 & 0 \\ f_{21} & 1 & 0 & 0 & 0 & 0 \\ f_{31} & f_{32} & 1 & f_{34} & f_{35} & 0 \\ f_{41} & f_{42} & f_{43} & 1 & f_{45} & 0 \\ f_{51} & f_{52} & 0 & f_{54} & 1 & 0 \\ f_{61} & f_{62} & f_{63} & f_{64} & f_{65} & 1 \end{bmatrix}
\begin{vmatrix} \mu_t^{COMM\_PRICES} \\ \mu_t^{SA\_REPO} \\ \mu_{it}^{RGDP\_G} \\ \mu_{it}^{INF} \\ \mu_{it}^{MS} \\ \mu_{it}^{LRATE} \end{vmatrix}
=
\begin{bmatrix} b_1 & 0 & 0 & 0 & 0 & 0 \\ 0 & b_2 & 0 & 0 & 0 & 0 \\ 0 & 0 & b_3 & 0 & 0 & 0 \\ 0 & 0 & 0 & b_4 & 0 & 0 \\ 0 & 0 & 0 & 0 & b_5 & 0 \\ 0 & 0 & 0 & 0 & 0 & b_6 \end{bmatrix}
\begin{vmatrix} \varepsilon_t^{COMM\_PRICES} \\ \varepsilon_t^{SA\_REPO} \\ \varepsilon_{it}^{RGDP\_G} \\ \varepsilon_{it}^{INF} \\ \varepsilon_{it}^{MS} \\ \varepsilon_{it}^{LRATE} \end{vmatrix} \quad (6)
$$

The first matrix at the left-hand side refers to the model's non-recursive restrictions, whilst the matrix on the right-hand side denotes the diagonal matrix. Furthermore, the reduced-form residuals are expressed as follows: $\mu_t^{COMM\_PRICES}$, $\mu_t^{SA\_REPO}$, $\mu_{it}^{RGDP\_G}$, $\mu_{it}^{INF}$, $\mu_{it}^{MS}$, and $\mu_{it}^{LRATE}$, which are disturbances to domestic and foreign variables in the CMA region. They represent unexpected movements depending on the information given on each variable in the structure. Moreso, the structural shocks in the equations are denoted with the following expressions $\varepsilon_t^{COMM\_PRICES}$, $\varepsilon_t^{SA\_REPO}$, $\varepsilon_{it}^{RGDP\_G}$, $\varepsilon_{it}^{INF}$, $\varepsilon_{it}^{MS}$, and $\varepsilon_{it}^{LRATE}$. Whereas the expressions $(f_{21} - f_{65})$ included in the matrices above are non-zero coefficients which reflect the instantaneous effect of variables, the sluggish response is denoted by 0.

### 4.2. Model Specification

Adopting the study of Kutu and Ngalawa (2016), commodity prices entered the panel-SVAR as an external shock to mitigate the price puzzle problem, and in line with Seleteng (2016) the anchor country's repo rate enters the panel SVAR as a second exogenous variable. Endogenous variables such as money supply, lending rates, real GDP growth, and inflation were included as economic performance indicators in line with Ikhide and Uanguta (2010) and Seleteng (2016).

### 4.3. Data

The study employed annual data for the period 1980 to 2021. The data for commodity prices were obtained from the Quantec database. Data for each country's repo rate and performance variables were obtained from World Bank Development Indicators website, Federal Reserve Bank of St Louis (FRED), and reports from the Central Bank of Lesotho, Central bank of Eswatini, Bank of Namibia and Reserve of South Africa.

All variables except the SA repo rate are in annual frequencies. A frequency conversion method in EViews was used to change the repo rate from monthly to annual frequencies. This is consistent with the study of Kutu and Ngalawa (2016). Following Berkelmans' (2005) study, which supports using data in levels rather than differencing, unit roots tests were not conducted to avoid loss of vital information. Hence, the panel-SVAR was estimated in levels. A lag length of 4 was chosen in line with Kato (2019), and block exogeneity tests were conducted following Sokhanvar (2019) study.

### 4.4. Shocks Identification

Following the method coined by Amisano and Giannini (1997); Akande (2017) zero restrictions are imposed on *E* and *V* matrices. Furthermore, on both *E* and *V* matrices, the scheme needs 51 restrictions or $2n^2 - n(n+1)/2$, whereby the number of variables is denoted as n. Moreso, 30 exclusion restrictions are applied on diagonal matrix *V*, while for the model to be just identified, 21 restrictions are applied on matrix *E*.

## 5. Estimation Results and Discussion

### 5.1. Pre-Estimation Tests Results

An optimal lag length of four (4) was selected based on SC, FPE, LR, AIC, and HQ methods, as shown in Table 1 below. It was also guided by a previous study by Kato (2019), which revealed that a lag length of four yields more accurate results without compromising the degrees of freedom.

The next step was to conduct diagnostic tests, which helped validate the findings to avoid mis-specifying the model's functional form. The normality tests result based on Jarque–Bera, Skewness, and Kurtosis tests in Table 2 below revealed that the residuals are normally distributed at the joint and individual level at 5% significance level. Hence the study did not reject the null hypothesis that residuals are normally distributed.

Moreover, the serial correlation test and heteroscedasticity test results in Tables 3 and 4 below show no autocorrelation and no heteroscedasticity problems in the model at a 5% significance level.

**Table 1.** Panel-SVAR Lag Length Test.

| Lag | LogL | LR | FPE | AIC | SC | HQ |
|-----|------|-----|-----|-----|-----|-----|
| 0 | 197.3920 | NA | $2.82 \times 10^{-9}$ | −2.658222 | −2.534480 | −2.607941 |
| 1 | 927.5774 | 1389.381 | $1.84 \times 10^{-13}$ | −12.29969 | −11.43349 | −11.94771 |
| 2 | 1096.575 | 307.4821 | $2.90 \times 10^{-14}$ | −14.14688 | −12.53823 | −13.49321 |
| 3 | 1228.303 | 228.6946 | $7.72 \times 10^{-15}$ | −15.47643 | −13.12533 | −14.52108 |
| 4 | 1588.826 | 595.8636 * | $8.62 \times 10^{-17}$ * | −19.98369 * | −16.89014 * | −18.72665 * |

* Indicates lag order selected by the criterion; LR: sequential modified LR test statistic (each test at 5% level); FPE: Final prediction error; AIC: Akaike information criterion; SC: Schwarz information criterion; HQ: Hannan–Quinn information criterion.

**Table 2.** Normality tests of the Panel-SVAR.

| Com | Skewness | | | | Kurtosis | | | | Jarque–Bera | | |
|-----|------|--------|----|-------|----------|--------|----|-------|--------|----|-------|
| | Skew | Chi-sq | Df | Prob | Kurtosis | Chi-sq | Df | Prob | Chi-sq | Df | Prob |
| 1 | 0.0749 | 8.0213 | 1 | 0.3247 | 4.7432 | 5.4356 | 1 | 0.8294 | 7.5438 | 2 | 0.7382 |
| 2 | −4.0987 | 3.5021 | 1 | 0.9823 | 6.7854 | 5.1245 | 1 | 0.6959 | 8.7659 | 2 | 0.3521 |
| 3 | 0.6548 | 2.6749 | 1 | 0.4325 | 8.7630 | 6.2321 | 1 | 0.2998 | 6.8765 | 2 | 0.7694 |
| 4 | −3.7654 | 3.2325 | 1 | 0.6329 | 2.6574 | 2.4327 | 1 | 0.9432 | 8.3478 | 2 | 0.5439 |
| 5 | −1.8765 | 3.5237 | 1 | 0.7322 | 4.2109 | 8.2319 | 1 | 0.7320 | 6.7845 | 2 | 0.3098 |
| 6 | −3.0789 | 6.4122 | 1 | 0.9654 | 2.5470 | 3.2359 | 1 | 0.8120 | 5.3298 | 2 | 0.2754 |
| Joint | | 4.6544 | 6 | 0.2134 | | 3.9750 | 6 | 0.5450 | 7.3294 | 12 | 0.4329 |

**Table 3.** Serial Correlation LM Test.

| Null: There Is No Serial Correlation | | |
|-----|-----|-----|
| Lags | LM-Stat | Prob |
| 1 | 132.7654 | 0.2978 |
| 2 | 320.7021 | 0.3290 |
| 3 | 129.5780 | 0.4329 |
| 4 | 321.2134 | 0.5482 |
| 5 | 128.6520 | 0.7429 |
| 6 | 273.4376 | 0.4321 |

**Table 4.** Heteroscedasticity test.

| Null: There Is No Heteroscedasticity | | |
|-----|-----|-----|
| Chi-sq | Df | Prob. |
| 5738.214 | 412 | 0.7542 |

In the block exogeneity test, the null hypotheses stating that there is no Granger causality when economic growth (RGDP_G) is the dependent variable was rejected (Table 5). Hence, the result portrays that in the CMA region, both exogenous (SA repo rate and commodity prices) and other endogenous variables such as lending rates, money supply, and inflation Granger cause variations in economic growth (RGDP_G).

**Table 5.** Block Exogeneity Tests.

| | Block Exogeneity Wald Tests | | | |
| --- | --- | --- | --- | --- |
| | Null: There Is No Granger Causality | | | |
| | Dependent Variable: RGDP_G | | | |
| **Excluded** | **Chi-sq** | **Df** | **Prob** | **Decision** |
| COMM_PRICES | 45.53247 | 3 | 0.0001 | Null rejected |
| SA_REPO | 32.32897 | 3 | 0.0000 | Null rejected |
| INF | 158.8709 | 3 | 0.0000 | Null rejected |
| MS | 62.21874 | 3 | 0.0000 | Null rejected |
| LRATE | 18.82309 | 3 | 0.0005 | Null rejected |
| All | 238.1297 | 15 | 0.0002 | Null rejected |

*5.2. Impulse Response Functions (IRFs)*

Therefore, the results in Figure 7 reveal that economic growth (RGDP_G) is negatively influenced after a one-standard-deviation shock is applied to the South African repo rate (SA_REPO) or SA Repo Rate shock. Graphically there was a contraction in economic growth (RGDP_G) as it gradually declined in the negative zone for the rest of the periods. This finding of the initial decline in RGDP_G after a shock or increase in the short-term interest rate is in line with Ayopo et al. (2016). The empirical deduction from this result reveals that monetary policy shocks from the South African repo rate in the CMA region negatively impact the region's economic growth.

Therefore, the study bridges the knowledge gap in past studies, such as Ikhide and Uanguta (2010), by incorporating the real GDP growth in a panel data environment. Moreover, after a shock or innovation is introduced to the South African repo rate, a sharp decline in inflation (INF) was noticed in the initial periods. It bottoms out from the fourth period and picks up before the impulse tapers off. This finding conforms to the a priori expectations which assert that restrictive monetary policy action decreases the general price level (Seleteng 2016). Hence, the CMA economies can successfully utilise the South African repo rate as an instrument to alleviate inflation.

Furthermore, after a one-standard-deviation shock was introduced to the South African repo rate money supply of CMA countries declines gradually in the initial periods and flattens for the remaining periods. This initial finding is in line with general expectations, which posit that there is a negative relationship between interest rates and money (Seleteng 2016). This finding fills the knowledge gap in the CMA region since the results of the responses of the money supply from past studies are inconclusive. For example, after a shock in the repo rate, Seoela's (2020) study discovered that the money supply in Lesotho increases sharply. Whilst Masha et al. (2007) revealed no significant influence on money supply after an innovation or shock is applied to the South African repo rate.

Lastly, as depicted in Figure 7 after a one-standard-deviation shock is applied, the impulse response function on CMA economies' lending rates (LRATE) initially shows a sharp increase at the beginning periods, which gradually declines thereafter and dies off for the remaining periods. This sharp increase at the starting periods is in line with the general expectations and the a priori, which states that banks react by increasing lending rates after a restrictive monetary policy (Masha et al. 2007).

The results in Figure 8 show the contemporaneous impact of the pooled money supply of all CMA economies on other variables. The money supply is treated as an endogenous target variable which is influenced indirectly by monetary authorities. Furthermore, the block exogeneity tests show that money supply is crucial in influencing economic growth (RGDP_G). The result in Figure 8 reveals that following a shock in the supply of money in CMA economies there is an initial increase in economic growth. This initial slight increase

in economic growth is in line with the a priori, which assert that expansionary monetary policies cause growth in output (Dillner 2021).

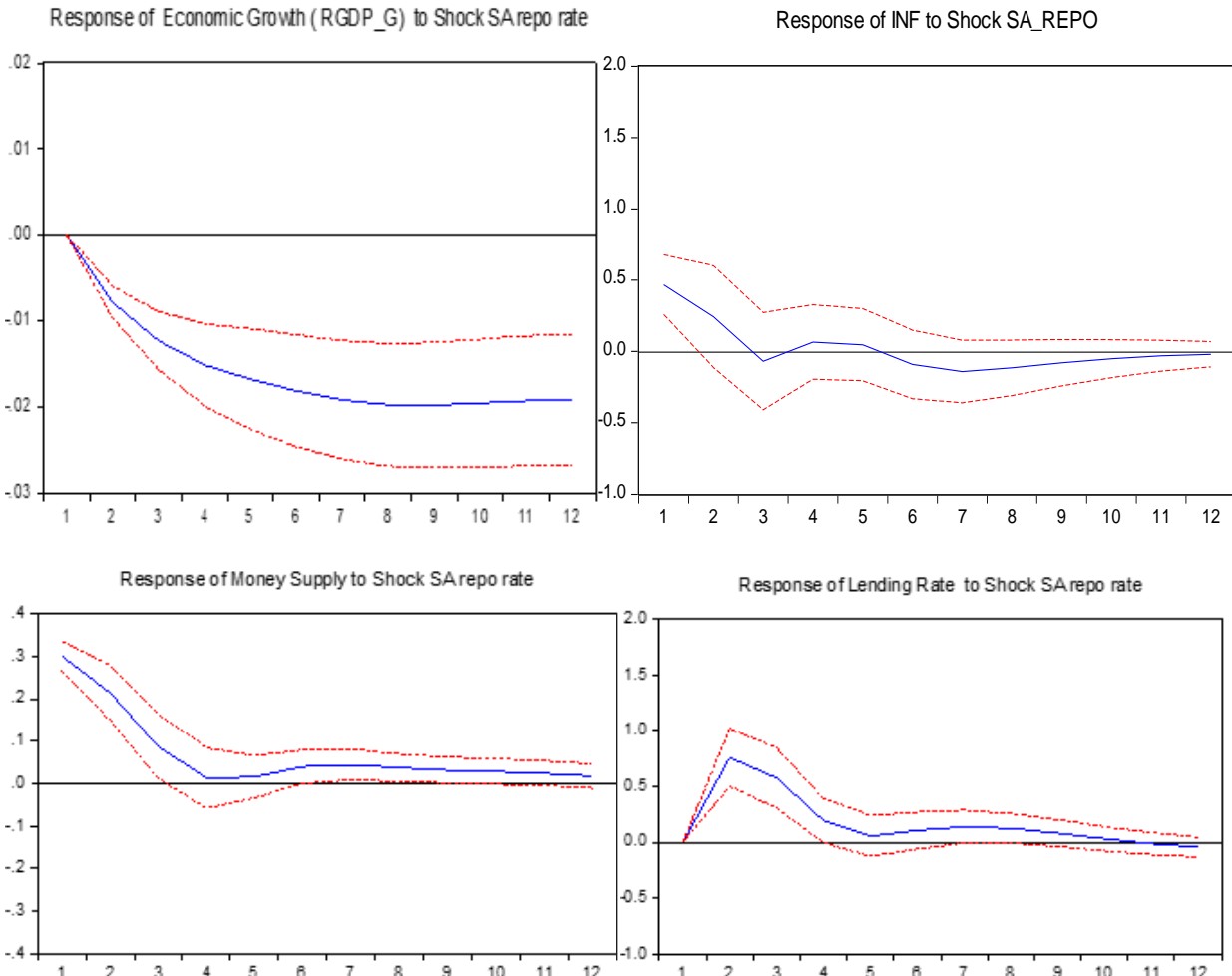

**Figure 7.** Impulse responses of Domestic Variables to Monetary Policy Shocks. The redlines are 95% confidence bands which give us certainty coming to the direction of change, and the blue line is the impulse response function.

Moreover, innovations to the supply of money initially causes a sharp hike in the general price levels in the CMA region. The increase of inflation at the beginning periods before it gradually declines is in line with empirical expectations, asserting that price levels rise after a loose monetary policy or when money supply is increased (Dillner 2021). Finally, Figure 8 portrays that after structural innovations on supply of money in the CMA, the lending rates decline at the earlier stages. This result is in line with the theoretical assertion which coins that there is an inverse relationship between money supply and short-term interest rates (Mohr and Fourie 2011). This implies that an increase in money supply will decrease lending rates and vice versa. These findings conform to Seleteng (2016).

### 5.3. Results of Variance Decomposition Analysis

This test result was presented to highlight a shock's significance in portraying the variables' variations. The variance decomposition denotes the percentage or proportion of innovation or shock on a particular variable associated with shock on other variables or a shock on itself or its own over a specific time period (Seleteng 2016). In conformity with the impulse response functions, the study incorporated twelve periods since yearly or annual data was employed. The 12 years or periods were divided into four-time or

four-year quarters. The first three quarters denote the short run, whereas the last peg or quarters demarcate the long run.

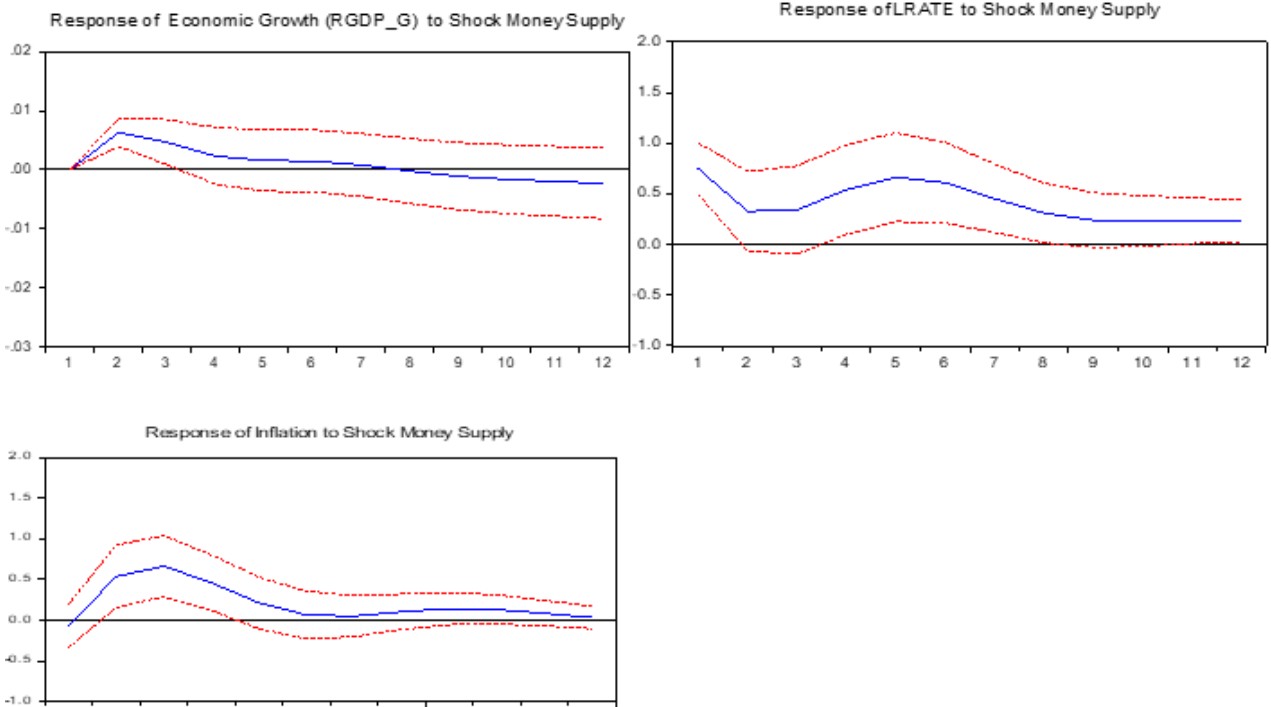

**Figure 8.** Impulse responses of Domestic Variables to Money Supply Shocks. The redlines are 95% confidence bands which give us certainty coming to the direction of change, and the blue line is the impulse response function.

Firstly, in Table 6 the results for the variance decompositions of commodity prices (a), therefore, reveals that commodity prices accounted for the significant contribution to its own shocks, whereas tiny contributions were accounted for from other variables. Particularly, 93.065%, 91.643%, 90.267%, and 90.013% in periods one, two, three, and four, were attributed to commodity prices' own shock, respectively. This finding conforms to Berkelmans' (2005) idea that shocks are transmitted from the global market to the CMA economy and not the other way around. This is connected to the fact that CMA region is still a small, open economy which is weak enough to be influenced by the global market.

Moreso, the variance decomposition of the South African repo rate (SA_REPO) denoted as (b), reveals that in the beginning quarter or period, the contribution of economic growth (RGDP_G), inflation, and lending rates shocks were not significant enough to account for changes in the repo rate. Together, 3.110% is attributed to commodity prices and money supply, whereas approximately 96.5389% is attributed to own shocks in causing changes in the SA repo rate in the first period. In the second to fourth quarters, the SA repo rate's own shocks still accounted for much of the variations. The marginal contribution of commodity prices of 1.480% in the first period, and 1.659%, 1.456%, and 1.861%, also from the second to the fourth periods in that order, shows that external shocks cannot affect the monetary policy tool of the anchor economy. This corresponds with the a priori expectations as SA_REPO is exogenously determined by the SARB.

Moreover, the variance decomposition of economic growth (RGDP_G) depicted as (c), pictures that much of the variations in RGDP_G are attributed to the SA repo rate. Particularly, from the first to the fourth quarter the SA repo rate accounted for 40.784%, 52.143%, 50.027%, and 46.638% in influencing changes in economic growth in the CMA economy. The empirical deduction from this result reveals that the SA repo rate is an

effective monetary instrument which can cause significant changes in economic growth in the CMA.

**Table 6.** Variance Decomposition Functions.

| Period | Standard Error | COMM_PRICES | SA_REPO | RGDP_G | INF | MS | LRATE |
|---|---|---|---|---|---|---|---|
| **(a) Variance decomposition of COMM_PRICES;** | | | | | | | |
| 3 | 0.127 | 93.065 | 1.387 | 0.882 | 1.538 | 0.155 | 2.973 |
| 6 | 0.148 | 91.643 | 3.239 | 1.723 | 1.653 | 0.573 | 1.169 |
| 9 | 0.172 | 90.267 | 1.564 | 3.521 | 3.143 | 0.255 | 1.250 |
| 12 | 0.177 | 90.013 | 2.126 | 2.124 | 2.537 | 0.674 | 2.526 |
| **(b) Variance decomposition of SA_REPO;** | | | | | | | |
| 3 | 0.212 | 1.480 | 96.538 | 0.159 | 0.166 | 1.630 | 0.027 |
| 6 | 0.344 | 1.659 | 91.978 | 0.762 | 1.288 | 1.733 | 2.580 |
| 9 | 0.472 | 1.456 | 90.179 | 1.538 | 1.376 | 1.439 | 4.012 |
| 12 | 0.585 | 1.861 | 88.364 | 2.176 | 2.004 | 1.173 | 4.422 |
| **(c) Variance decomposition of RGDP_G;** | | | | | | | |
| 3 | 0.571 | 6.217 | 40.784 | 42.829 | 4.349 | 2.277 | 3.544 |
| 6 | 0.594 | 7.370 | 52.143 | 32.689 | 2.569 | 1.533 | 3.696 |
| 9 | 0.691 | 7.179 | 50.027 | 32.289 | 3.886 | 2.573 | 4.046 |
| 12 | 0.888 | 7.632 | 46.638 | 36.446 | 2.348 | 2.883 | 4.053 |
| **(d) Variance decomposition of INF;** | | | | | | | |
| 3 | 0.847 | 12.422 | 11.378 | 7.861 | 62.009 | 3.651 | 2.679 |
| 6 | 0.328 | 24.829 | 6.986 | 6.143 | 56.153 | 2.222 | 3.667 |
| 9 | 0.986 | 28.898 | 7.828 | 5.879 | 51.621 | 2.184 | 3.590 |
| 12 | 0.587 | 29.976 | 7.975 | 5.544 | 50.549 | 2.765 | 3.191 |
| **(e) Variance decomposition of MS** | | | | | | | |
| 3 | 0.076 | 1.027 | 37.602 | 8.231 | 5.828 | 35.988 | 11.324 |
| 6 | 0.950 | 2.486 | 37.002 | 8.244 | 5.769 | 34.042 | 12.457 |
| 9 | 0.574 | 3.575 | 38.246 | 5.873 | 5.984 | 35.768 | 10.554 |
| 12 | 0.682 | 3.273 | 37.824 | 7.829 | 8.724 | 32.873 | 9.477 |
| **(f) Variance decomposition of LRATE** | | | | | | | |
| 3 | 0.247 | 8.853 | 76.231 | 0.678 | 0.787 | 2.861 | 10.590 |
| 6 | 0.548 | 9.207 | 73.647 | 1.748 | 3.472 | 1.219 | 10.707 |
| 9 | 0.576 | 9.631 | 66.482 | 2.032 | 5.465 | 2.808 | 13.582 |
| 12 | 0.583 | 9.826 | 67.233 | 2.535 | 5.932 | 4.624 | 9.850 |

Both money supply (MS) and lending rates (LRATE) shocks indicated tiny contributions in explaining the variations in RGDP_G. For instance, 2.277%, 1.533%, 2.573%, and 2.883% of variations in economic growth are attributed to money supply from the first quarter to the last quarter in that order. Whereas 3.544%, 3.696%, 4.046%, and 4.053% of variations in RGDP_G are accounted for by lending rates. Furthermore, 4.349%, 2.569%, 3.886%, and 2.348% of changes in RGDP_G are accounted for by inflation from the first quarter to the last in that order. On the other hand, Commodity prices (COMM_PRICES), in the first quarter contributed 6.217%, whilst 7.370%, 7.179%, and 7.632% from the second period to the fourth period, respectively, were attributed to causing changes in RGDP_G. This finding

corresponds with the result obtained from the IRFs in on the impact of COMM_PRICES on RGDP_G. This major contribution is highly expected since global or external shocks are considered to affect the small CMA economy and not the reverse. Therefore, as a policy recommendation in the short run, the CMA authorities must be careful and cautious of global shocks and implement economic policies that can cushion the negative effects of external shocks and also shocks from the SA repo rate. These findings add to previous studies in the CMA by Seoela (2020), Ikhide and Uanguta (2010), and Seleteng (2016), which did not incorporate the external effect of shocks such as global commodity prices in the CMA.

Table 6 further shows the variance decompositions of inflation (INF) identified as (d). It indicates that all variables account for causing much change in the inflation of CMA countries. Specifically, much of the variation in inflation is attributed to the SA repo rate and commodity prices, besides their own shock. For instance, from the first to the last period, commodity prices accounted for 12.422%, 24.829%, 28.898%, and 29.976% of variation, whilst 11.378%, 6.986%, 7.828%, and 7.975% is attributed to the SA repo rate in that order.

Moreover, Table 6 also reveals the variance decompositions of money supply (MS) labelled as (e). The findings showed that much change in money supply is attributed to the SA repo rate followed by its own shocks. Specifically, the SA repo rate accounts for 37.602%, 37.002%, 38.246%, and 37.824% of changes in money supply from the first to the last period in that order. Whereas money supply's own shocks accounted for 35.988%, 34.042%, 35.768%, and 32.873% of influence on its own changes. This implies that the SA repo rate is an important tool which causes changes or reduces the supply of money in the CMA as shown in the IRFs graph above.

Finally, Table 6 denotes the variance decomposition of lending rates depicted as (f). The findings show that major contributions to LRATE fluctuations were accounted for from the SA repo rate. Particularly, the SA repo rate accounted for 76.231%, 73.647%, 66.482%, and 67.233% of fluctuations from the first to the final period in that order. This finding is in complete harmony with the IRFs results. This is also consistent with the empirical findings of Ikhide and Uanguta (2010), that the repo rate is the most appropriate monetary policy tool to effect economic changes in the CMA.

## 6. Conclusions and Recommendation

The study employed the panel-SVAR model utilising IRFs, and variance decompositions to achieve its primary objective of assessing the impact of monetary policy shocks from the leading economy on the CMA region's economic activity. Conclusively, the study revealed that a shock from the South African repo rate or the main monetary instrument of the anchor country decreases inflation, reduces output growth, increases lending rates, and reduces the money supply in the CMA region. Based on the findings of the study it is suggested that policy makers and monetary authorities in the CMA should look inward and formulate policies towards stimulating the output of the entire region in order to offset or circumvent worsening economic performance in the CMA region.

Comparatively, although these findings were similar to the international study of Cavallo and Ribba (2015), which revealed that monetary policy shocks result in recessionary effects in the Euro Area. However, in the CMA, unlike in previous studies, global commodity prices were incorporated in this study as a global control variable which captures external factors since the CMA countries interact with other countries in the world market. Our findings further buttress that global commodity price as an exogenous variable has a forecasting ability which also helps in mitigating the price puzzle problem. Sims (1992), the first author to comment on the empirical anomaly of the price puzzle in VARs which is associated with monetary tightening, also verified the use of commodity prices to arrest the price puzzle problem. Hence, by including global commodity prices in the SVAR this study differs from both international and national studies. Hence this study also bridged the literature gap in solving the unconventional result of the price puzzle

connected with restrictive or tight monetary policy. Global commodity prices can also serve as an information variable which can help the SARB in setting policy rates accurately as opined by the proponents of the Taylor rule. This study therefore fills the literature gap by including global commodity prices which adds onto Ikhide and Uanguta (2010), Seleteng (2016), and Seoela (2020), whose studies did not incorporate the global variable. Finally, it is highly recommended that governmental authorities in the CMA region establish a safety net and implement policies capable of stimulating economic growth and averting poor growth. Such policies must also be able to cushion the effect of unexpected shocks from the SA monetary policy decisions and also global shocks that may hit the region of the CMA. Hence, the CMA economies are urged to deliberate on the establishment of a common pool of reserves which will assist some of its members who form part of their integrated group and might need assistance due to shocks that might have hit their economies.

The research on regional integration must be ongoing since it is at the forefront of many countries. In the CMA region, aspects such as the inclusion of the SACU revenue in analysing the spillover effects of the leading economy's monetary policy shock and external shocks remain uncovered. This is because the CMA countries form part of the SACU countries, and the revenue from the SACU may act as a cushion to galvanise small countries from potential shocks.

**Author Contributions:** Conceptualization, T.S.; methodology, T.S.; software, T.S.; validation, T.S. and S.M.; formal analysis, T.S.; investigation, T.S.; resources, T.S.; data curation, T.S.; writing—original draft preparation, T.S.; writing—review and editing, T.S.; visualization, T.S.; supervision, S.M.; project administration, T.S.; funding acquisition, T.S. and S.M. All authors have read and agreed to the published version of the manuscript.

**Funding:** This research received no external funding.

**Informed Consent Statement:** Not applicable.

**Data Availability Statement:** The authors greatly appreciate the datasets from the World Bank Development Indicators (https://databank.worldbank.org/source/world-development-indicators/, retrieved on 9 August 2022); Datasets from the Federal Reserve Bank of st Louis (https://fred.stlouisfed.org/series/INTDSRZAM193N, retrieved on 3 August 2022); datasets from Quantec data base (https://www.quantec.co.za/easydata/, accessed on 7 June 2022); Monthly statistical release from the Central Bank of Eswatini (https://www.centralbank.org.sz/monthly-statistical-releases/, accessed on 5 August 2022); data sets from the Central Bank of Lesotho (https://www.centralbank.org.ls/index.php/statistics, accessed on 7 August 2022); datasets from the Bank of Namibia (https://www.bon.com.na/, accessed on: 9 September 2022); data sets from Stats SA (Statistics South Africa) (https://www.statssa.gov.za/?page_id=1854&PPN=P0141/, accessed on 1 July 2022); and data sets from the South African Reserve Bank (https://www.resbank.co.za/en/home/publications/quarterly-bulletin1/download-information-from-xlsx-data-files/, accessed on 20 July 2022).

**Conflicts of Interest:** The authors declare no conflict of interest.

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
