# Peer review of "Monetary Policy Implications on Macroeconomic Performance in the Common Monetary Area: A Panel-SVAR Framework"

_economies, doi:10.3390/economies11050144_

Round 1

Reviewer 1 Report

Summary

This paper studies the dynamic effects of monetary policy shocks from South Africa on a set of macroeconomic variables of the Common Monetary Area. This Area includes South Africa, Namibia, Lesotho and Eswatini. The author(s) estimate and identify a Structural Panel VAR over the sample 1980 – 2021. The main conclusion is that contractionary monetary shocks originating from South Africa exert recessionary effects on the whole Area.  

Comments

I believe that the paper deals with an interesting subject and uses an appropriate methodology, based on Panel-SVAR. Nonetheless, in my opinion there are some problems and shortcomings that need to be tackled before publication. In particular, I have found some inconsistencies both in part concerning the empirical results and in the related macroeconomic interpretation. Below some specific comments are proposed.  

1.     The review of the relevant literature might be improved, in particular in relation to empirical investigations conducted with alternative methodologies and concerning other economic areas. Two important papers, among others, might be considered: the first is the paper by Cavallo and Ribba, “Common Macroeconomic Shocks and Business Cycle Fluctuations in Euro Area Countries”, International Review of Economics and Finance, 38 (2015), 377-392, that uses the near-VAR methodology; the second is the paper by Georgiadis, “Determinants of Global Spillovers from US Monetary Policy”, Journal of International Money and Finance, 67 (2016), 41-61, that uses a Global VAR model.

2.     The authors show the effects of monetary policy shocks from South Africa on real GDP and inflation in the Common Monetary Area. However, it would be of interest to obtain the impulse response functions concerning the member countries. Indeed, the overall results characterizing the Area might hide different response of the national macroeconomic variables.  

3.     I have found some inconsistencies in the interpretation of the results. It seems to me that sometimes the authors confuse the sign of the dynamic responses of some variables with the shape charactering the responses themselves. For example, as far as the response of inflation to a contractionary monetary policy shock is concerned, the conclusion is wrong. Since, it is worth noting that inflation increases in response to an unexpected increase of the interest rate in South Africa. Thus, there is evidence of an inflation puzzle and the authors should discuss and explain this wrong sign affecting inflation.  

4.     It is a common practice in structural VAR analysis to undertake some robustness check of the results obtained. For example, this robustness analysis is presented in the above cited paper by Cavallo and Ribba (2015), who identify first the Euro-area monetary policy shock by imposing contemporaneous restrictions and then evaluate the robustness of the results by identifying the structural shocks by imposing sign restrictions on the response of some variables. In the light of the inflation puzzle shown in the present study, that after all might be ascribed to misspecification of the model and/or to problems related to the identification strategy adopted, I believe that such robustness analysis should be undertaken.

Author Response

Responses to the comments of Reviewer 1

  1. Comment 1: The relevant literature has been added. See the track changes in the main manuscript.
  2. Comment 2: The study used annual data from 1980 to 2021. The major drawback of time series data, in this case, is few observations, especially for an SVAR framework. This was one of the setbacks mentioned in Seoela (2020)’s study which used time series data. Hence employing time series data in order to obtain individual member countries' impulse response functions would mean few degrees of freedom which yields results with less reliable statistical inference. The choice for panel data is justifiable in our study as it gives our model more degrees of freedom and is more efficient than time series data since it prevents collinearity of variables (Gujarati and Porter, 2009). Pooling cross-sectional data together has the advantage of dynamic adjustment and computing the effects that are not visible in time series (Famoroti and Tipoy, 2019). Another advantage of employing panel data is that it prevents the loss of information. This is because panel data is suitable for a group-wise or regional analysis rather than an individual-based investigation (Kim, 2017; Monfort et al., 2003).
  3. Moreover, employing panel data is preferable compared to time-series data because in a panel data framework, the total number of observations increases. Incorporating panel data also reduces the noise which comes from the individual time series; heteroscedasticity is not an issue in panel data analysis. Peersman and Smets (2001) assert that in a situation where data is not enough for individual analysis, especially in developing countries where data availability is an issue and not enough to fit regressions which use time series data, panel data analysis is suitable. Finally, panel estimation techniques take into consideration the heterogeneity associated with individual data.
  4. Comment 3:On the response of Inflation to Shock 2 (SA-REPO), there was a typo mistake. The author erroneously extracted the wrong impulse response graph from the full sample results. The correct graph has been extracted and replaced. For verification purposes, the author has included full sample results of the impulse response functions. Column two shows all the IRFs after a one standard shock was introduced to the SA- REPO. Please see the track changes.
  5. Comment 4: The full sample results obtained reflect no evidence of price puzzle. In our opinion, there is no need to conduct robustness tests since there is no price puzzle in the model. Since these are not estimates, according to Ngalawa and Viegi (2011), variables are bound to revert to their “unconditional means” after a shock. The confidence bands give us certainty coming to the direction of change. In this instance, the initial response from the first period reflected by the impulse response graph shows that inflation is decreasing.

Reviewer 2 Report

After analyzing the article I have the following recommendations in order to improve the manuscript:

1. In the Introduction Section, it must be inserted the main added value of the paper. Moreover, in the introductory Section, it must be highlighted the scope and Objective of the research as well as the actuality and importance of the research theme.

2. The last Paragraph of the Introduction Section must be the ”road map” of the article. To be more specific the structure of the article is in the flow of the article.

3. In the Section ”economic performance in the CMA” the Author(s) need to clarify how they defined the economic performances and how is this applied to a specific Area like CMA? This must be well structured and explained.

4. In the Section ”monetary policy shocks and economic performance” since it is the Literature review section that presents the current state of the art it must be inserted a table which must encompass the empirical national and international Studies which deal with the problem and the Method and outcomes they have arrived. Critical Analysis is needed here!

5. In the conclusions and recommendations Section, the Author(s) must insert the following elements: (i) discussion regarding the findings vis-a-vis other Studies Developed by the authors at national and international levels and their differences?! (ii) what is the strong point of the findings in this paper which contributes to the current state of the art?! (iii) what are the limitations of the Study?! (iv) what are the following paths of future research that Will be developed by the Author(s) and the concrete directions?!

6. The Reference List must be improved Both by classical Studies in the Field But also by actual international Studies published in recognized journals worldwide.

Author Response

Response to reviewer 2

.

Comment 1

The changes have been implemented in the introduction.

 Previous studies in South Africa reveal that periods of restrictive monetary policy have a negative effect on output growth in South Africa specifically industrial output (Kutu and Ngalawa, 2016; Kabundi and Ngwenya, 2011; Seleteng, 2016; Ikhide and Uanguta , 2010). However, very few studies have extended their analysis to the CMA region. There is still a lack of clarity on how monetary decisions from South Africa can influence the CMA region and the transmission channels through which the effects are conveyed.

Please see the track changes in the manuscript.

Comment 2:

The last Paragraph of the Introduction has been amended see track changes in the main manuscript.

The objective of the study is focused on evaluating monetary policy induced innovations and how they affect macroeconomic performance indicators in the region. This study closes the literature gap by comparatively analysing the transmission of monetary policy shocks in the CMA.

Comment 3:

. Economic performance clearly defined.

Economic performance can be defined as a measurement or indicator of how well an economy is doing in achieving its most crucial objectives (Kashima, 2017). According to Chileshe et al. (2018) the most predominant and key economic objectives targeted by any economy are price stability and stable high rate of economic growth. The main objective behind the establishment of the CMA monetary union is in fostering economic advancement and development of less developed participant members (Jian-Ye et al., 2006).  In the measurement of economic performance economic growth, the Real Gross Domestic Product Growth (RGDP_G) rate has been regarded as a prominent indicator in the CMA region (Kashima, 2017). RGDP_G reveals the annual percentage change in the total value of goods and services produced in an economy adjusted for inflation (Kaboro et al., 2018).

Please see track changes in the main manuscript

Comment 4:

Other international studies forexample Cavallo and Ribba (2015 and   Georgiadis (2016), Buigut (2009) have been added. The others observed that there is no consensus in the literature body, for instance Ikhide and Uanguta (2010) found out that a shock in the repo rate decreases prices or inflation in CMA countries, whereas the results of Dlamini (2018) and Seoela (2020) reflect the opposite.

The most notable study of Cavallo and Ribba (2015)  using a structural (near) VAR investigated the impact of area wide shocks with particular attention to monetary policy shocks. Their conclusion was that a contractionary monetary policy cause similar recessionary effects in all countries and that as far as business cycle fluctuations are concerned, the largest European economies were mainly explained by common area-wide shocks, whereas the second category of economies comprising Portugal, Ireland and Greece national shocks played a greater role.

Moreso, Georgiadis (2016), employed a Global VAR (GVAR) model to assess the global spillovers from identified US monetary policy shocks. The study found out that US monetary policy generates sizable output spillovers to the rest of the world, which are larger than the domestic effects in the US for many economies. The results suggested that policy makers could mitigate their economies’ vulnerability to US monetary policy by fostering trade integration as well as domestic financial market development, increasing the flexibility of exchange rates, and reducing frictions in labour markets.

Buigut (2009), estimated a three-variable recursive VAR for three East African Community (EAC) three countries using data from 1984 and 2006. The paper found that a shock to the short-term interest rate was found to have no statistical significance effect on real output and inflation. Nevertheless, Kutu and Ngalawa (2016) argued that these findings are biased by the fact that the study used a sample that includes too few observations for empirical analyses, resulting in few degrees of freedom. Kutu and Ngalawa (2016)) proposed that using a panel-SVAR model could resolve these issues because it provides an effective way of dealing with over-parameterization. In contrast, their results found that an expansionary monetary increases prices significantly in Kenya and Uganda, while output increases in Burundi, Kenya, and Rwanda. Similar to the current study, there are very few studies that compare the effect of South Africa’s monetary policy conduct on the performance of the CMA countries. Ikhide and Uanguta (2010) and Seleteng (2016) both examined the impact of SARB’s monetary policy on the CMA economies using a VAR framework. Unlike the approach in this study, Ikhide and Uanguta (2010) used monthly data but excluded economic output from the estimated models, while Seleteng (2016) used annually aggregated data. These studies focused on how changes in the SARB’s monetary policy instrument (repo rate) affect the money supply, credit, and prices in the CMA and thus evaluate the ability of the CMA economies to undertake independent monetary policy. Both studies found statistically significant results that lending rates and price levels were instantaneously sensitive to changes in the repo rate. However, Ikhide and Uanguta (2010) also found that money supply is instantaneously responsive to the repo rate, while Seleteng (2016) did not find any significant relationship.

Overall, the previous studies show opposing views on the impact of monetary policy on economic performance. Some support that increasing interest rates (contractionary monetary policy) negatively affects the economy, while others support that it has no observable impact on the economy

Comment 5:

  1. The conclusions and recommendations have been amended in line with the reviewer’s suggestions. Conclusively, the study revealed that a shock from the main monetary instrument of the anchor country decreases inflation, reduces output growth, increases lending rates, and reduces the money supply in the CMA region. Comparatively, although these findings were similar to the international study of Cavallo and Riba (2016), which revealed that monetary policy shocks result in recessionary effects in the Euro Area. However, in the CMA region unlike the notable study of Ikhide and Uanguta (2010) which did not incorporate the output variable, this study included real GDP which is regarded as a prominent performance indicator. Moreover, in the CMA this study differs from the study of Seoela (2020), whose findings revealed that a shock in the SA_REPO increases inflation in all CMA countries which contradicts theory. Seoela (2020) identified their result as an anomaly known as the price puzzle, which pointed out that the policymakers could be setting the interest rate without observing future inflation signals (Rossouw et al., 2014). Despite that Seoela’s (2020) study did not incorporate global commodity prices. Other authors suggested that the inclusion of this variable in SVAR models assists in curbing the price puzzle (Kamati, 2020; Canova, 2011).

Giordani (2004) also asserts that the forescasting ability of global commodity prices helps to mitigate the price puzzle. Hence, by including global commodity prices in the SVAR this study differs from both international and national studies. Hence this study also bridged the literature gap identified in Seoela (2020)’s unconventional result after monetary tightening. Global commodity prices can also serve as an information variable which can help the SARB in setting policy rates accurately as opined by the proponents of the Taylor rule (Giordani, 2004). Sims (1992), the first author to comment on the empirical anomaly in VARs which is associated with monetary tightening, verified the use of commodity prices to arrest the price puzzle problem. This study, therefore, fills the literature gap by including global commodity prices which adds to Ikhide and Uanguta (2010) and Seoela (2020), whose studies did not incorporate the global variable.  Hence, it is highly recommendable that governmental authorities in the CMA region establish a safety net and implement policies capable of stimulating economic growth and averting poor growth. Such policies must also be able to cushion the effect of unexpected shocks from the SA monetary policy decisions and also global shocks that may hit the region of the CMA.

 The research on regional integration must be ongoing since it is at the forefront of many countries. In the CMA region, aspects such as the inclusion of the SACU revenue in analysing the spillover effects of the leading economy's monetary policy shock and external shocks remain uncovered. This is because the CMA countries form part of the SACU countries and the revenue from the SACU may act as a cushion to galvanise small countries from potential shocks.

Please see the track changes in the revised main manuscript.

Comment 6:

The author has improved the reference list by adding some international studies such as Georgiadis (2016);  Buigut (2009); Cavallo and Ribba (2015) and others.  Please see the track changes in the main manuscript.

Round 2

Reviewer 1 Report

I believe that the author(s), in this revised version, have sufficiently addressed the concerns expressed in my previous report. Nevertheless, in my opinion, some further revisions are needed in the last section, Conclusion and Recommendation. Indeed, the section is affected by a number of typos, e.g. "Cavallo and Riba (2016)" instead of Cavallo and Ribba (2015), "Giordani (2004) also asserts the forescasting ability...". More generally, this section should include some conclusions and policy implications rather than another round of literature review.  

Author Response

The typo errors have been amended eg the reference now reads Cavallo and Ribba (2015). The section now includes conclusions and policy recommendations as suggested by the reviewer.

Please see the track changes which reads:

 “Conclusively, the study revealed that a shock from the South African repo rate or the main monetary instrument of the anchor country decreases inflation, reduces output growth, increases lending rates, and reduces the money supply in the CMA region. Based on the findings of the study it is suggested that policy makers and monetary authorities in the CMA should look inward and formulate policies towards stimulating the output of the entire region in order to   offset or circumvent worsening economic performance in the CMA region.

Comparatively, although these findings were similar to the international study of Cavallo and Ribba (2015), which revealed that monetary policy shocks result in recessionary effects in the Euro Area. However, in the CMA unlike previous studies global commodity prices were incorporated in this study as a global control variable which captures external factors since the CMA countries interact with other countries in the world market.  Our findings further buttress that global commodity prices as an exogenous variable have a forecasting ability which also helps in mitigating the price puzzle problem. Sims (1992), the first author to comment on the empirical anomaly in VARs which is associated with monetary tightening, also verified the use of commodity prices to arrest the price puzzle problem. Hence, by including global commodity prices in the SVAR this study differs from both international and national studies. Hence this study also bridged the literature gap in solving the unconventional result of the price puzzle connected with restrictive or tight monetary policy. Global commodity prices can also serve as an information variable which can help the SARB in setting policy rates accurately as opined by the proponents of the Taylor rule. This study therefore fills the literature gap by including global commodity prices which adds onto Ikhide and Uanguta (2010), Seleteng (2016) and Seoela (2020), whose studies did not incorporate the global variable.  Finally, it is highly recommendable that governmental authorities in the CMA region establish a safety net and implement policies capable of stimulating economic growth and averting poor growth. Such policies must also be able to cushion the effect of unexpected shocks from the SA monetary policy decisions and also global shocks that may hit the region of the CMA.  Hence, the CMA economies are urged to deliberate on the establishment of a common pool of reserves which will assist some of its members who form part of their integrated group and might need assistance due to shocks that might have hit their economies’’.

Reviewer 2 Report

The Authors improve their manuscript which has a better Quality now.

Author Response

Thank you for your comments which improved my paper and the compliments on the last review changes.